microbiology

*Wolbachia*, mosquitoes, malaria, *Anopheles*, endosymbionts

**Author for correspondence:**
Thomas Walker
e-mail: thomas.walker@lshtm.ac.uk

†Present address: Departments of Vector Biology and Tropical Disease Biology, Centre for Neglected Tropical Diseases, Liverpool School of Tropical Medicine, Liverpool, UK.

A contribution to the Life Sciences New Talent special collection.

# Evidence for natural hybridization and novel *Wolbachia* strain superinfections in the *Anopheles gambiae* complex from Guinea

Claire L. Jeffries[1], Cintia Cansado-Utrilla[1,†],
Abdoul H. Beavogui[2], Caleb Stica[1], Eugene K. Lama[3],
Mojca Kristan[1], Seth R. Irish[4] and Thomas Walker[1]

[1]Department of Disease Control, Faculty of Infectious and Tropical Diseases, London School of Hygiene and Tropical Medicine, London WC1E 7HT, UK
[2]Centre National de Formation et de Recherche en Santé Rurale de Mafèrinyah B.P. 2649, Conakry, Guinea
[3]Programme National de Lutte contre le Paludisme, Guinée, B.P. 6339 Conakry, Guinea
[4]The US President's Malaria Initiative and Entomology Branch, Centers for Disease Control and Prevention, Atlanta, GA 30329-4027, USA

CLJ, 0000-0002-0298-2808; TW, 0000-0002-3545-012X

*Wolbachia*, a widespread bacterium which can influence mosquito-borne pathogen transmission, has recently been detected within *Anopheles* (*An.*) species that are malaria vectors in Sub-Saharan Africa. Although studies have reported *Wolbachia* strains in the *An. gambiae* complex, apparent low density and prevalence rates require confirmation. In this study, wild *Anopheles* mosquitoes collected from two regions of Guinea were investigated. In contrast with previous studies, RNA was extracted from adult females ($n = 516$) to increase the chances for the detection of actively expressed *Wolbachia* genes, determine *Wolbachia* prevalence rates and estimate relative strain densities. Molecular confirmation of mosquito species and *Wolbachia* multilocus sequence typing (MLST) were carried out to analyse phylogenetic relationships of mosquito hosts and newly discovered *Wolbachia* strains. Strains were detected in *An. melas* (prevalence rate of 11.6%–16/138) and hybrids between *An. melas* and *An. gambiae* sensu stricto (prevalence rate of 40.0%–6/15) from Senguelen in the Maferinyah region. Furthermore, a novel high-density strain, termed *w*AnsX, was

found in an unclassified *Anopheles* species. The discovery of novel *Wolbachia* strains (particularly in members, and hybrids, of the *An. gambiae* complex) provides further candidate strains that could be used for future *Wolbachia*-based malaria biocontrol strategies.

# 1. Introduction

*Wolbachia* endosymbiotic bacteria are estimated to infect approximately 40% of insect species [1] and natural infections have been shown to have inhibitory effects on human arboviruses in mosquitoes [2–4]. High-density *Wolbachia* strains have been used for mosquito biocontrol strategies targeting arboviruses as they induce synergistic phenotypic effects. *Wolbachia* strains that have been transinfected into *Aedes* (*Ae.*) *aegypti* and *Ae. albopictus* induce inhibitory effects on arboviruses, with maternal transmission and cytoplasmic incompatibility (CI) enabling introduced strains to spread through populations [5–13]. The successful release and establishment of *Wolbachia*-transinfected *Ae. aegypti* populations in Cairns, Australia [14] was followed by further evidence of strong inhibitory effects on arboviruses from field populations [15]. Further studies in Australia [16,17] and Kuala Lumpur, Malaysia [18] have now shown that *Wolbachia* frequencies have remained stable since initial releases, and there is a reduction in human dengue incidence (case notifications) in the release sites.

The potential for *Wolbachia* to be used for biocontrol strategies targeting malaria transmission by *Anopheles* (*An.*) species has also been postulated [19] and initial laboratory experiments demonstrated that transient infections in *An. gambiae* reduce the density of *Plasmodium* (*P.*) *falciparum* parasites [20]. However, as with arboviruses, there is variability in the level of inhibition of malaria parasites for different *Wolbachia* strains in different mosquito species [21–23]. A major step forward was achieved through the transinfection of a *Wolbachia* strain from *Ae. albopictus* (*w*AlbB) into *An. stephensi* and the confirmation of *P. falciparum* inhibition [24]. The interest in using *Wolbachia* for biocontrol strategies targeting malaria transmission in *Anopheles* mosquitoes has further increased due to the detection of natural strains of *Wolbachia* residing in numerous malaria vectors of Sub-Saharan Africa [25–29]. The *An. gambiae* complex, which consists of multiple morphologically indistinguishable species including several major malaria vector species, appears to contain diverse *Wolbachia* strains (collectively named *w*Anga) at both low prevalence and low infection densities [25,26,28–31]. As the majority of studies have used nested-PCR for detection, more robust evidence is required to determine whether *Wolbachia* strains are established as endosymbionts in *Anopheles* species [31]. The majority of these studies are limited to amplification of only a few genes (particularly *16S rRNA*), and this is problematic given the possibility of amplifying prokaryotic *16S rRNA* genes from non-living cells [32]. By contrast, the recently discovered *w*AnM and *w*AnsA strains, found in *An. moucheti* and *An.* species A, respectively, are higher density infections that dominate the mosquito microbiome [26,33].

Interestingly, the presence of *Wolbachia* strains in *Anopheles* was inversely correlated to other bacterial species such as *Asaia* that are stably associated with several species [34–37]. Evidence for this 'mutual exclusion' between bacterial species in *Anopheles* was also present from analysis of field-collected mosquitoes from multiple countries in Sub-Saharan Africa [26]. In this study, we collected wild *Anopheles* mosquitoes from two regions of Guinea in June–July 2018 and characterized the natural *Wolbachia* strains to provide further evidence for the presence of these endosymbionts in malaria vectors. In contrast with previous studies, we extracted RNA to make any detection of *Wolbachia* more likely to be from actively expressed *Wolbachia* genes and undertook qRT-PCR analysis to compare *Wolbachia* densities. Phylogenetic analysis revealed the presence of novel strains in *An. melas*, *An. gambiae* sensu stricto (s.s.)–*melas* hybrids (including *Wolbachia* superinfections within individual mosquitoes) and an unclassified *Anopheles* species.

# 2. Material and methods

## 2.1. Study sites and collection methods

*Anopheles* adult mosquitoes were collected in 2018 from two regions (sub-prefectures) in Guinea: Faranah and Maferinyah. Human landing catches (HLCs) and larval dipping were conducted in three villages in the Faranah Prefecture: Balayani (10.1325, −10.7443), Foulaya (10.144633, −10.749717) and Tindo (9.9612230, −10.7016560) [38]. Three districts were selected for mosquito collections in the Maferinyah

sub-prefecture using a variety of traps [39]. BG-Sentinel 2 traps (BG2) (Biogents), CDC light traps (John W. Hock), gravid traps (BioQuip) and stealth traps (John W. Hock) were used to sample adult mosquitoes in Maferinyah Centre I (9.54650, −13.28160), Senguelen (9.41150, −13.37564) and Fandie (9.53047, −13.24000). Mosquitoes collected from traps and HLCs were morphologically identified using keys and stored in RNAlater® (Invitrogen) at −70°C [38,39].

## 2.2. RNA extraction and generation of complementary DNA

RNA was extracted from individual whole female mosquitoes using Qiagen 96 RNeasy Kits according to manufacturer's instructions and a Qiagen Tissue Lyser II (Hilden, Germany) with a 5 mm stainless steel bead (Qiagen) to homogenize mosquitoes. RNA was eluted in 45 µl of RNase-free water and stored at −70°C. RNA was reverse transcribed into complementary DNA (cDNA) using an Applied Biosystems High Capacity cDNA Reverse Transcription kit. A final volume of 20 µl contained 10 µl RNA, 2 µl 10X RT buffer, 0.8 µl 25X dNTP (100 mM), 2 µl 10X random primers, 1 µl reverse transcriptase and 4.2 µl nuclease-free water. Reverse transcription was undertaken in a Bio-Rad T100 Thermal Cycler as follows: 25°C for 10 min, 37°C for 120 min and 85°C for 5 min and cDNA stored at −20°C.

## 2.3. Molecular mosquito species identification

Species identification of the *An. gambiae* complex was initially undertaken using diagnostic species-specific PCR assays targeting the ribosomal intergenic spacer (IGS) [40] and SINE200 insertion [41] to distinguish between the morphologically indistinguishable sibling species. To confirm species identification for samples of interest and samples that could not be identified by species-specific PCR, Sanger sequencing and phylogenetic analysis was performed for PCR products from a range of gene targets including ribosomal IGS and internal transcribed spacer 2 (ITS2) [42] and mitochondrial cytochrome c oxidase subunit 1 (*COI*) [43], cytochrome c oxidase subunit 2 (*COII*) [44] and NADH dehydrogenase subunits 4 and 5 (*ND4-ND5*) [45]. Where ITS2 PCR products for a particular sample were not successfully generated, or the sequencing generated was not of sufficient quality for onward analysis, a slight modification to the ITS2 primers was used to attempt to increase the success of amplification and sequencing. Alternative ITS2 primers adjusted from those published [42] were ITS2A-CJ: 5′-TGTGAACTTGCAGGACACAT-3′ and ITS2B-CJ: 5′-TATGCTTAAATTYAGGGGGT-3′. For confirmation of *Culex (Cx.) watti*—a species collected in the same location and used for comparative *Wolbachia* density analysis—a different fragment of the mitochondrial cytochrome c oxidase subunit 1 (*COI*) gene [46] was sequenced given the lack of available sequences in certain regions for this species and to optimize sequencing quality and species discrimination. PCR reactions for IGS, SINE200, ITS2 and *COI* were prepared as previously described [39]. For *COII* amplification, PCR reactions were prepared using 10 µl of Phire Hot Start II PCR Master Mix (Thermo Scientific™) with a final concentration of 1 µM of each primer, 1 µl of PCR grade water and 2 µl template cDNA, to a final reaction volume of 20 µl. PCR reactions were carried out in a Bio-Rad T100 Thermal Cycler and cycling was 98°C for 30 s followed by 34 cycles of 98°C for 5 s, 55°C for 5 s, 72°C for 30 s followed by 72°C for 1 min. For *ND4-ND5* PCR, reactions were prepared using 10 µl of HotStart Taq 2x Master Mix (New England BioLabs®) with a final concentration of 2 µM of each primer, 1 µl of PCR grade water and 2 µl template cDNA, to a final reaction volume of 20 µl. PCR reactions were carried out in a Bio-Rad T100 Thermal Cycler and cycling was 95°C for 30 s followed by 35 cycles of 95°C for 30 s, 53°C for 60 s, 68°C for 90 s followed by 68°C for 5 min. PCR products were separated and visualized using 2% E-Gel EX agarose gels (Invitrogen) with SYBR safe and an Invitrogen E-Gel iBase Real-Time Transilluminator.

## 2.4. *Wolbachia* detection and amplification of *Wolbachia* genes

*Wolbachia* detection was first undertaken on cDNA targeting the conserved *Wolbachia* genes previously shown to amplify a wide diversity of strains; *16S rRNA* gene using primers W-Spec-16S-F: 5′-CATACCTATTCGAAGGGATA-3′ and W-Spec-16s-R: 5′-AGCTTCGAGTGAAACCAATTC-3′ [47] and *Wolbachia* surface protein (*wsp*) gene using primers wsp81F: 5′-TGGTCCAATAAGTGATGAAGAAAC-3′ and wsp691R: 5′-AAAAATTAAACGCTACTCCA-3′ [48]. Multilocus strain typing (MLST) was undertaken to characterize *Wolbachia* strains using the sequences of five conserved genes as molecular markers to genotype each strain. In brief, 450–500 base pair fragments of the *gatB, coxA, hcpA, ftsZ* and *fbpA Wolbachia* genes were amplified from individual *Wolbachia*-infected mosquitoes using previously optimized protocols [49,50]. Primers used were as follows: gatB_F1: 5′-GAKTTAAAYCGYGCAGGBGTT-

3′, gatB_R1: 5′-TGGYAAYTCRGGYAAAGATGA-3′, coxA_F1: 5′-TTGGRGCRATYAACTTTATAG-3′, coxA_R1: 5′-CTAAAGACTTTKACRCCAGT-3′, hcpA_F1: 5′-GAAATARCAGTTGCTGCAAA-3′, hcpA_R1: 5′-GAAAGTYRAGCAAGYTCTG-3′, ftsZ_F1: 5′-ATYATGGARCATATAAARGATAG-3′, ftsZ_R1: 5′-TCRAGYAATGGATTRGATAT-3′, fbpA_F1: 5′-GCTGCTCCRCTTGGYWTGAT-3′ and fbpA_R1: 5′-CCRCCAGARAAAAYYACTATTC-3′ with the addition of M13 adaptors. If no amplification was detected using standard primers, further PCR analysis was undertaken using degenerate primer sets, with or without M13 adaptors [49]. In selected *An. melas* specimens where *Wolbachia 16S rRNA* Sanger sequencing (detailed below) indicated the possibility of superinfections, further MLST testing was carried out using *Wolbachia* Supergroup A and B strain-specific primers [49]. PCR reactions were prepared using 10 µl of Phire Hot Start II PCR Master Mix (Thermo Scientific™) with a final concentration of 1 µM of each primer, 1 µl of PCR grade water and 2 µl template cDNA, to a final reaction volume of 20 µl. PCR reactions were carried out in a Bio-Rad T100 Thermal Cycler using variable optimized cycling conditions. For *gatB*, *hcpA* and *fbpA* genes, cycling was 98°C for 30 s followed by 34 cycles of 98°C for 5 s, 65°C for 5 s, 72°C for 10 s followed by 72°C for 1 min. For *coxA* and *ftsZ* genes, cycling was 98°C for 30 s followed by 34 cycles of 98°C for 5 s, 55°C for 5 s and 72°C for 30 s followed by 72°C for 1 min. PCR products were separated and visualized using 2% E-Gel EX agarose gels (Invitrogen) with SYBR safe and an Invitrogen E-Gel iBase Real-Time Transilluminator.

## 2.5. Sanger sequencing

PCR products were submitted to Source BioScience (Source BioScience Plc, Nottingham, UK) for PCR reaction clean-up, followed by Sanger sequencing to generate both forward and reverse reads. Where *Wolbachia* PCR primers included M13 adaptors, just the M13 primers alone (M13_adaptor_F: 5′-TGTAAAACGACGGCCAGT-3′ and M13_adaptor_R: 5′-CAGGAAACAGCTATGACC-3′) were used for sequencing; otherwise, the same primers as used for PCR were used. Sequencing analysis was carried out in MEGAX [51]. Both chromatograms (forward and reverse traces) from each sample were manually checked, edited and trimmed as required, followed by alignment by ClustalW and checking to produce consensus sequences. Consensus sequences were used to perform nucleotide BLAST (NCBI) database queries and for *Wolbachia* genes searches against the *Wolbachia* MLST database (https://pubmlst.org/organisms/wolbachia-spp/). If a sequence produced an exact match in the MLST database we assigned the appropriate allele number; otherwise, we obtained a new allele number for each novel gene locus sequence for *Anopheles Wolbachia* strains through submission of the FASTA and raw trace files on the *Wolbachia* MLST website for new allele assignment and inclusion within the database. Full consensus sequences were also submitted to GenBank and assigned accession numbers. The Sanger sequencing traces from the *wsp* gene were also treated in the same way and analysed alongside the MLST gene locus scheme, as an additional marker for strain typing. Where potential mixed strains were detected (in *An. melas* and *An. gambiae* s.s.–*melas* hybrid individuals) and any further Supergroup A or B specific testing was exhausted, it was not possible to submit these sequences to the MLST database for a new allele to be assigned; however, clean 16S consensus sequences from representative individuals for each of the Supergroup A and B strains characterized were submitted to GenBank, in addition to the full MLST profile of one individual demonstrating one of the Supergroup A strain infections.

## 2.6. Phylogenetic analysis

Alignments were constructed in MEGAX by ClustalW to include all relevant and available sequences highlighted through searches on the BLAST and *Wolbachia* MLST databases. Maximum-likelihood (ML) phylogenetic trees were constructed from Sanger sequences as follows. The most appropriate nucleotide substitution model for each phylogenetic analysis was selected through the use of the 'Find Best-Fit Substitution Model (ML)' option within the MEGAX software. The model with the lowest Bayesian information criterion (BIC) score from this analysis is considered to describe the substitution pattern the best. Options to model non-uniformity of evolutionary rates among sites using a discrete Gamma distribution (+G) with five rate categories and by assuming that a certain fraction of sites is evolutionary invariable (+I) were also evaluated during this analysis to highlight the most appropriate model and options to use for construction of each phylogenetic tree. The evolutionary history was then inferred by using the ML method with the most appropriate model and options for each respective tree selected, with details of the methods used for each specific tree included in the figure legends. The models used in the analysis included the Jukes–Cantor model [52], the Kimura two-parameter model [53], the general time reversible model [54], the Hasegawa–Kishino–Yano model [55] and the Tamura three-parameter model

[56]. The tree with the highest log likelihood in each case is shown. The percentage of trees in which the associated taxa clustered together is shown next to the branches. The phylogeny test was by the bootstrap method with 1000 replications. Initial tree(s) for the heuristic search were obtained automatically by applying neighbour-join and BioNJ algorithms to a matrix of pairwise distances estimated using the maximum composite likelihood (MCL) approach, and then selecting the topology with superior log likelihood value. The trees are drawn to scale, with branch lengths measured in the number of substitutions per site. Codon positions included were 1st + 2nd + 3rd + Noncoding. All positions containing gaps and missing data were eliminated. Evolutionary analyses were conducted in MEGAX [51].

## 2.7. *Wolbachia* quantification

To estimate *Wolbachia* density across multiple mosquito species, RNA extracts were added to Qubit™ RNA High-Sensitivity Assays (Invitrogen) and total RNA measured using a Qubit 4 Fluorometer (Invitrogen). All RNA extracts were then diluted to produce extracts that were 2.0 ng µl$^{-1}$ prior to being used in quantitative reverse-transcription PCR (qRT-PCR) assays targeting the *Wolbachia 16S rRNA* gene [28]. A synthetic oligonucleotide standard (Integrated DNA Technologies) was designed to calculate *16S rRNA* gene copies per µl using a 10-fold serial dilution (electronic supplementary material, figure S1). *16S rRNA* gene real-time qRT-PCR reactions were prepared using 5 µl of QuantiNova SYBR® Green RT-PCR Kit (Qiagen), a final concentration of 1 µM of each primer, 1 µl of PCR grade water and 2 µl template RNA, to a final reaction volume of 10 µl. Prepared reactions were run on a Roche LightCycler® 96 System for 15 min at 95°C, followed by 40 cycles of 95°C for 15 s and 58°C for 30 s. Amplification was followed by a dissociation curve (95°C for 10 s, 65°C for 60 s and 97°C for 1 s) to ensure the correct target sequence was being amplified. Each mosquito RNA extract was run in triplicate alongside standard curves and no template controls (NTCs) and PCR results were analysed using the LightCycler® 96 software (Roche Diagnostics).

## 2.8. *Asaia* detection

*Asaia* PCR screening was undertaken by targeting the *Asaia 16S rRNA* gene using primers Asafor: 5′-GCGCGTAGGCGGTTTACAC-3′ and Asarev: 5′-AGCGTCAGTAATGAGCCAGGTT-3′ [35,57]. *Asaia 16S rRNA* gene real-time qRT-PCR reactions were prepared using 5 µl of QuantiNova SYBR® Green RT-PCR Kit (Qiagen), a final concentration of 1 µM of each primer, 1 µl of PCR grade water and 2 µl template DNA, to a final reaction volume of 10 µl. Prepared reactions were run on a Roche LightCycler® 96 System for 15 min at 95°C, followed by 40 cycles of 95°C for 15 s and 58°C for 30 s. Amplification was followed by a dissociation curve (95°C for 10 s, 65°C for 60 s and 97°C for 1 s) to ensure the correct target sequence was being amplified.

## 2.9. Statistical analysis

Normalized qRT-PCR *Wolbachia 16S rRNA* gene copies per µl were compared using unpaired *t*-tests in GraphPad Prism 7.

# 3. Results

## 3.1. Mosquito species and *Wolbachia* strain prevalence rates

In addition to confirmation of species for the morphologically indistinguishable individuals within the *An. gambiae* complex, initial screening using diagnostic species-specific PCRs highlighted the presence of some naturally occurring hybrids between members of the *An. gambiae* complex. Concomitant PCR screening demonstrated the presence of *Wolbachia* within individuals of the *An. gambiae* complex, including a number of the hybrid specimens (electronic supplementary material, table S1). The composition of these hybrids was further investigated and confirmed through a repeat of the normally multiplex ribosomal IGS PCR [40] in single-plex format, separating the *An. gambiae* s.s./ *coluzzii* primer set from the *An. melas* primer set, achieving strong amplification for both target sequences (figure 1*a*) and confirmed for some representative samples through Sanger sequencing and phylogenetic analysis of both IGS PCR products from the same individuals (figure 1*b*). The further use of PCR amplification, Sanger sequencing and phylogenetic analysis of the ribosomal ITS2

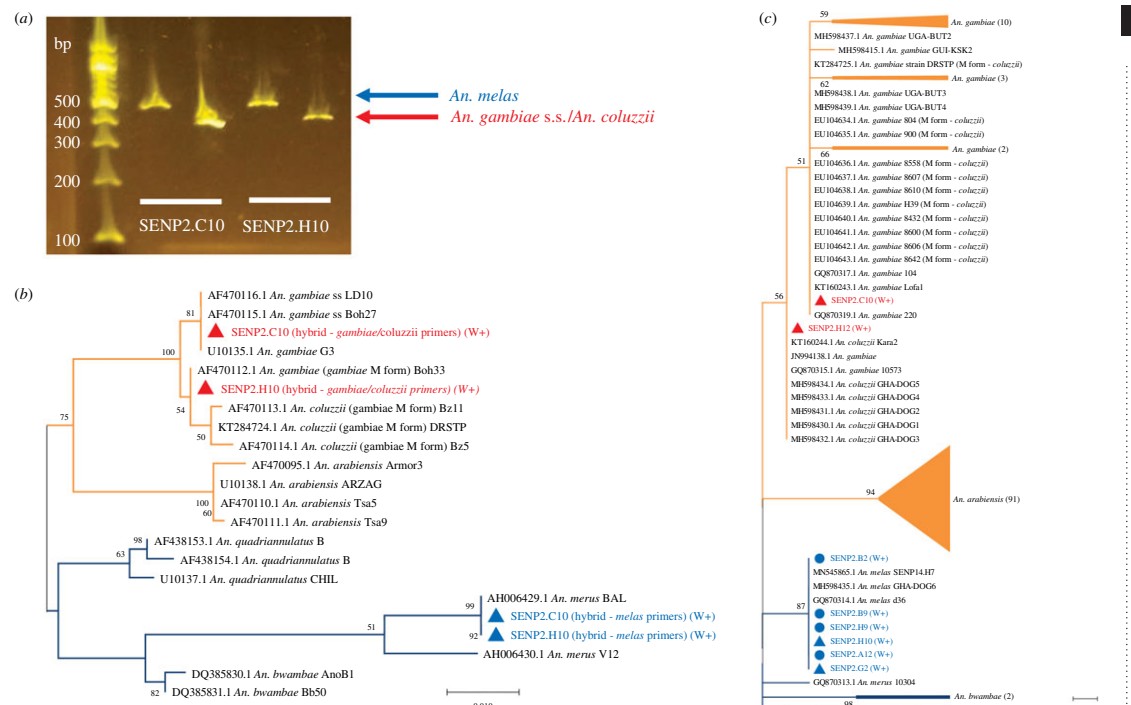

**Figure 1.** *Anopheles gambiae* complex PCR and phylogenetic analysis of the ribosomal IGS and ITS2 gene fragments. (*a*) Gel electrophoresis analysis of IGS *An. gambiae/melas* primer split down PCR products from two representative *Wolbachia* positive *An. gambiae* s.s.–*melas* hybrids. (*b*) Maximum-likelihood molecular phylogenetic analysis of sequences from IGS *An. gambiae/melas* primer split down PCR products for representative *Wolbachia* positive (W+) hybrid samples. Sequences from IGS *An. melas* specific primer set PCR products (blue) are shown alongside IGS *An. gambiae* primer set PCR products (red). The Jukes–Cantor model was used. A discrete Gamma distribution was used to model evolutionary rate differences among sites (five categories (+G, parameter = 0.6126)). The rate variation model allowed for some sites to be evolutionarily invariable ([+I], 44.56% sites). The tree with the highest log likelihood (−1952.82) is shown. The tree is drawn to scale, with branch lengths measured in the number of substitutions per site. The analysis involved 22 nucleotide sequences. There was a total of 901 positions in the final dataset. Sequences obtained from GenBank for comparison are shown with their accession numbers. (*c*) Maximum-likelihood molecular phylogenetic analysis of *An. gambiae* complex ITS2 sequences to demonstrate ribosomal ITS2 phylogeny of *Wolbachia* positive (W+) *An. melas* (blue circles) and *An. gambiae* s.s.–*melas* hybrids (blue and red triangles). The Kimura two-parameter model was used. The tree with the highest log likelihood (−1365.13) is shown. The tree is drawn to scale, with branch lengths measured in the number of substitutions per site. The analysis involved 149 nucleotide sequences. There were a total of 528 positions in the final dataset. Relevant subtrees are compressed, labelled with the species and the number of sequences included within them shown in brackets. Accession numbers are shown for sequences obtained from GenBank for comparison, where they are not contained within a subtree for clear visualization.

(figure 1*c*) and mitochondrial *COI, COII* (electronic supplementary material, figure S2) and *ND4-ND5* (electronic supplementary material, figure S3) genes was able to confirm both the mosquito species identity for individuals of interest and the composition of hybrids, with mitochondrial gene analysis indicating the maternal species identity. Prevalence rates of natural *Wolbachia* strains were variable depending on *Anopheles* species and location (table 1). *Wolbachia* strains were detected in *An. gambiae* s.s. mosquitoes from the Faranah region with prevalence rates ranging from 0.0 to 2.8%. In the Maferinyah region, from individuals collected in Senguelen, *Wolbachia* strains were detected in *An. melas* (11.6%—16/138) and in *An. gambiae* s.s.-*melas* hybrids (40.0% prevalence—6/15). Interestingly, *Wolbachia* was not found in any of the 4 *An. gambiae* s.s., 18 *An. coluzzii*, 2 *An. coluzzii–gambiae* s.s. hybrids or an *An. coluzzii-melas* hybrid collected from Senguelen, suggesting *Wolbachia* strains are not currently widespread across all members of the *An. gambiae* complex in this location. Phylogenetic hybrid composition analysis combined with *Wolbachia* screening highlighted the majority of *An. gambiae* s.s.–*melas* hybrids collected from Senguelen had *An. melas* mothers (8/12 *An. melas* by mitochondrial analysis), with 4/6 *Wolbachia* positive hybrids having *An. melas* as the maternal species. These results, combined with the prevalence of maternally inherited *Wolbachia* in the *An. melas* individuals and not in the *An. gambiae* s.s. individuals from this location, suggests the *Wolbachia* in this population has most likely originated from *An. melas*.

**Table 1.** *Wolbachia* prevalence rates in *Anopheles* species collected in two regions of Guinea in 2018. Species containing *Wolbachia*-infected individuals are denoted in bold.

| region | location | species | *Wolbachia*+ individuals | total individuals | prevalence (%) |
|---|---|---|---|---|---|
| **Faranah** | **Balayani** | ***An. gambiae*** s.s. | **4** | **143** | **2.80** |
| Faranah | Balayani | *An. coluzzii* | 0 | 1 | 0.00 |
| Faranah | Balayani | *An. coluzzii-gambiae* s.s. hybrid | 0 | 1 | 0.00 |
| Faranah | Balayani | species unknown | 0 | 1 | 0.00 |
| Faranah | Faranah | *An. gambiae* s.s. | 0 | 26 | 0.00 |
| Faranah | Faranah | *An. coluzzii* | 0 | 1 | 0.00 |
| Faranah | Foulaya | *An. gambiae* s.s. | 0 | 63 | 0.00 |
| Faranah | Foulaya | *An. coluzzii-gambiae* s.s. hybrid | 0 | 1 | 0.00 |
| **Faranah** | **Tindo** | ***An. gambiae*** s.s. | **1** | **48** | **2.08** |
| Faranah | Tindo | *An. coluzzii* | 0 | 1 | 0.00 |
| Faranah | Tindo | *An. coluzzii-gambiae* s.s. hybrid | 0 | 2 | 0.00 |
| Maferinyah | Fandie | *An. coluzzii* | 0 | 20 | 0.00 |
| Maferinyah | Fandie | *An. gambiae* s.s. | 0 | 1 | 0.00 |
| Maferinyah | Fandie | *An. melas* | 0 | 2 | 0.00 |
| Maferinyah | Fandie | *An. gambiae* s.s.-*melas* hybrid | 0 | 4 | 0.00 |
| Maferinyah | Fandie | *An. coustani* | 0 | 1 | 0.00 |
| Maferinyah | Maferinyah | *An. coluzzii* | 0 | 6 | 0.00 |
| Maferinyah | Maferinyah | *An. coustani* | 0 | 3 | 0.00 |
| Maferinyah | Maferinyah | *An. gambiae* s.s. | 0 | 1 | 0.00 |
| Maferinyah | Maferinyah | *An. coluzzii-gambiae* s.s. hybrid | 0 | 1 | 0.00 |
| Maferinyah | Maferinyah | *An. squamosus* | 0 | 8 | 0.00 |
| Maferinyah | Senguelen | *An. coluzzii* | 0 | 18 | 0.00 |
| Maferinyah | Senguelen | *An. coluzzii-melas* hybrid | 0 | 1 | 0.00 |
| Maferinyah | Senguelen | *An. coluzzii-gambiae* s.s. hybrid | 0 | 2 | 0.00 |
| **Maferinyah** | **Senguelen** | ***An. gambiae* s.s.-*melas* hybrid** | **6** | **15** | **40.00** |
| Maferinyah | Senguelen | *An. coustani* | 0 | 1 | 0.00 |
| Maferinyah | Senguelen | *An. gambiae* s.s. | 0 | 4 | 0.00 |
| **Maferinyah** | **Senguelen** | ***An. melas*** | **16** | **138** | **11.59** |
| **Maferinyah** | **Senguelen** | ***An.* species X** | **1** | **1** | **100.00** |
| Maferinyah | Senguelen | *An. squamosus* | 0 | 1 | 0.00 |

*Wolbachia*-negative *An. gambiae* s.s.–*melas* hybrids were also confirmed for two specimens from Fandie (with *An. gambiae* s.s. mitochondrial results) and a *Wolbachia*-negative *An. coluzzii*–*gambiae* s.s. hybrid (maternally *An. coluzzii*) from Maferinyah.

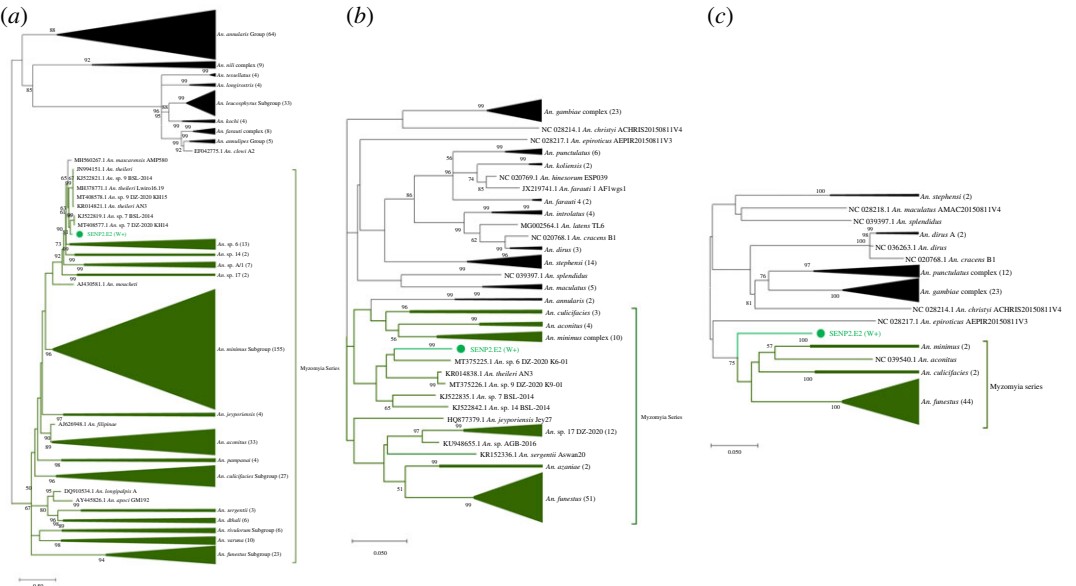

**Figure 2.** Phylogenetic analysis of the *An.* sp. X ribosomal ITS2, mitochondrial *COI* and *ND4-ND5* gene sequences within the *Cellia* subgenus of *Anopheles*. (*a*) Maximum-likelihood molecular phylogenetic analysis of ribosomal ITS2 sequences. All available GenBank sequences covering the sequenced fragment from the *Myzomyia*, *Neomyzomyia* and *Annularis* series were included, in addition to representative sequences from the *An. gambiae* complex (*Pyretophorus* Series) for broader placement and comparison with the other phylogenetic analyses. The general time reversible model was used. A discrete Gamma distribution was used to model evolutionary rate differences among sites (five categories (+G, parameter = 1.8538)). The rate variation model allowed for some sites to be evolutionarily invariable ([+I], 10.30% sites). The tree with the highest log likelihood (−32560.43) is shown. The analysis involved 440 nucleotide sequences. There was a total of 1112 positions in the final dataset. (*b*) Maximum-likelihood molecular phylogenetic analysis of mitochondrial *COI* sequences. All available GenBank sequences covering the sequenced fragment from the *Cellia* subgenus were included. The general time reversible model was used. A discrete Gamma distribution was used to model evolutionary rate differences among sites (five categories (+G, parameter = 0.3077)). The rate variation model allowed for some sites to be evolutionarily invariable ([+I], 33.43% sites). The tree with the highest log likelihood (−6158.48) is shown. The analysis involved 159 nucleotide sequences. There was a total of 658 positions in the final dataset. (*c*) Maximum-likelihood molecular phylogenetic analysis of mitochondrial *ND4-ND5* sequences. The Hasegawa–Kishino–Yano model was used. A discrete Gamma distribution was used to model evolutionary rate differences among sites (five categories (+G, parameter = 0.5879)). The rate variation model allowed for some sites to be evolutionarily invariable ([+I], 55.17% sites). All available GenBank sequences covering the sequenced fragment from the *Cellia* subgenus were included. The tree with the highest log likelihood (−9739.99) is shown. The analysis involved 95 nucleotide sequences. There was a total of 1518 positions in the final dataset. In all trees, the sequences generated in this study are shown in bold, with the *Wolbachia* positive *An.* sp. X specimen (SENP2.E2 (W+)) shown in green, with a filled circle node marker. Branches where sequences from the *Myzomyia* Series are grouping are shown in dark green, and with an external labelled dark green bracket to denote this grouping. Relevant subtrees are compressed, labelled with the species and the number of sequences included within them shown in brackets. Accession numbers are shown for sequences obtained from GenBank for comparison, where they are not contained within a subtree for clear visualization.

A *Wolbachia* strain was also found in a single female of an unclassified *Anopheles* species from Senguelen. Sanger sequencing and BLAST analysis of the ITS2 region revealed this *Anopheles* sp. 'X' was most similar to *Anopheles* sp. 7 BSL-2014 (GenBank accession number KJ522819.1) but at only 93.2% sequence identity, and *An. theileri* (GenBank accession number MH378771.1) with 90.9% sequence identity (both full query coverage). Phylogenetic analysis of the ribosomal ITS2 region and mitochondrial *COI* and *ND4-ND5* regions for *An.* sp. 'X' (figure 2) revealed that this species is from the Myzomyia Series, within the *Cellia* subgenus of *Anopheles*, with the agreement for this placement across all three phylogenies. The ITS2 region gave the greatest discrimination for this species; however, currently, no other sequences from this species are available in order to classify it any further than to Series level and closest to, but distinct from, sequences denoted *Anopheles* sp. 7, another as yet undetermined *Anopheles* species [58]. Mosquito ribosomal and mitochondrial gene sequences were deposited in GenBank and accession numbers obtained (electronic supplementary material, table S2).

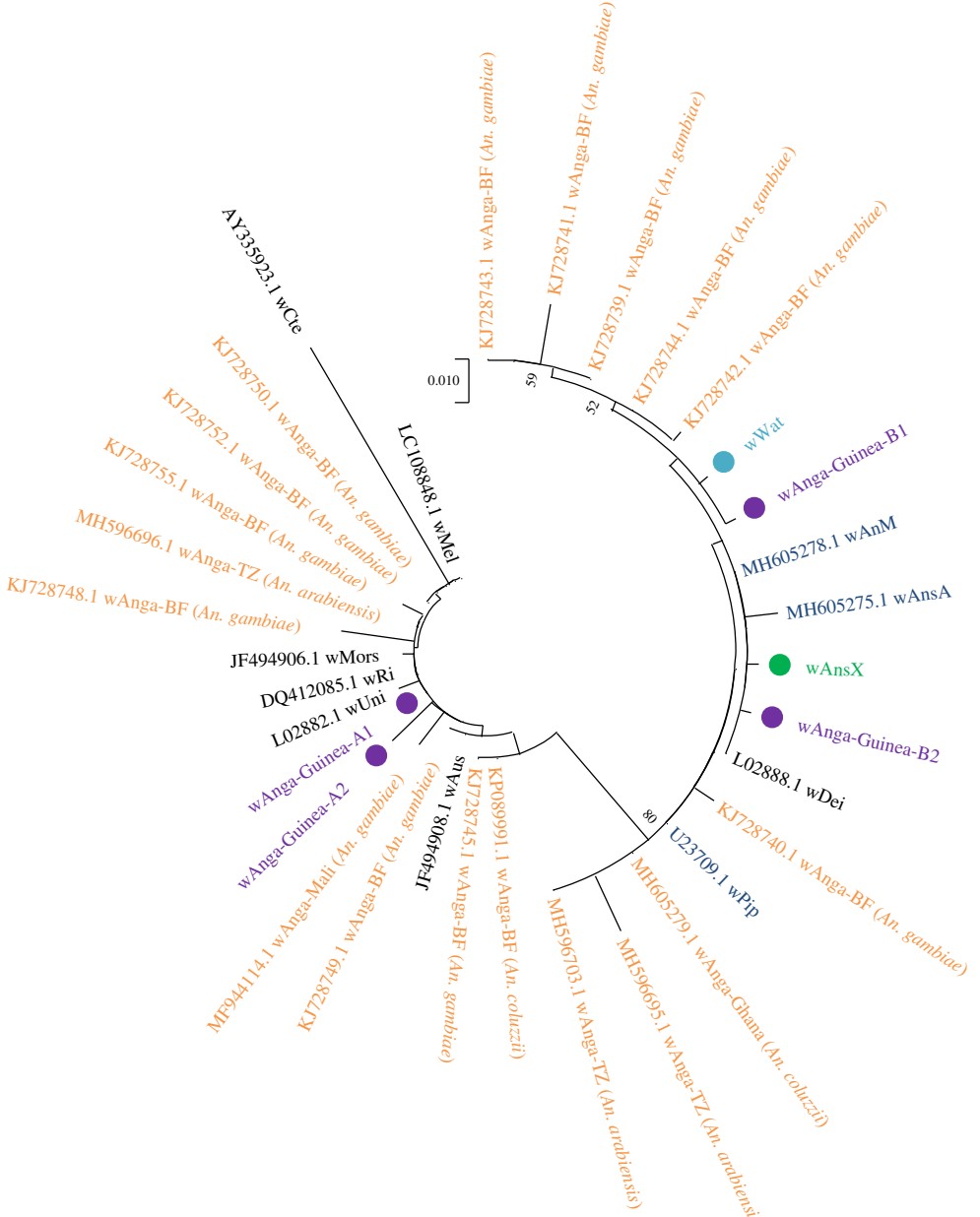

**Figure 3.** *Wolbachia* strain phylogenetic analysis using the *16S rRNA* gene. Maximum-likelihood molecular phylogenetic analysis of the *16S rRNA* gene for *Wolbachia* strains detected in *Anopheles* specimens from Guinea. The Kimura two-parameter model was used. A discrete Gamma distribution was used to model evolutionary rate differences among sites (five categories (+G, parameter = 0.4333)). The tree with the highest log likelihood (−1039.08) is shown. The tree is drawn to scale, with branch lengths measured in the number of substitutions per site. The analysis involved 34 nucleotide sequences. There was a total of 398 positions in the final dataset. Sequences generated in this study are shown in bold with filled circle node markers. The *w*Anga-Guinea sequences from both A and B Supergroups are shown in purple. The *w*AnsX strain is shown in green and the *w*Wat strain obtained from *Cx. watti* for comparative work is shown in light blue. *w*Anga sequences previously obtained from species in the *An. gambiae* complex are shown in orange with their accession numbers. *Wolbachia* strains obtained from other mosquito species in previous studies are shown in navy blue with their accession numbers. Additional *Wolbachia* sequences from non-mosquito hosts obtained from GenBank for comparison are shown with their accession numbers in black.

## 3.2. *Wolbachia* strain typing

Although amplification of the *Wolbachia 16S rRNA* fragments of the natural strain in *An. gambiae* s.s. from the Faranah region was possible, sequences obtained were of insufficient quality for further analysis. Furthermore, no *wsp* gene amplification was possible from *An. gambiae* s.s. from the Faranah region. By contrast, *Wolbachia 16S rRNA* (figure 3) and *wsp* sequences (electronic supplementary material, figure S4)

**Table 2.** Novel resident *Wolbachia* strain *WSP* typing and multilocus sequence typing (MLST) gene allelic profiles. Newly assigned novel alleles for *w*AnsX are shown in bold. *\*w*Anga-Guinea-A1 *hcpA* could not be assigned a novel allele number due to a possible double infection which was unresolvable, therefore the allele number of the closest match (CM) is shown with the number of single nucleotide differences to the closest match in brackets.

| mosquito species | *Wolbachia* strain | *WSP* typing allele numbers | | | | | MLST gene allele numbers | | | | |
| | | *wsp* | HVR1 | HVR2 | HVR3 | HVR4 | *gatB* | *coxA* | *hcpA* | *ftsZ* | *fbpA* |
| --- | --- | --- | --- | --- | --- | --- | --- | --- | --- | --- | --- |
| *An. melas* | *w*Anga-Guinea-A1 | 23 | 1 | 12 | 21 | 19 | 1 | 1 | CM1 (2)* | 3 | 1 |
| *An.* sp. X | *w*AnsX | **737** | **264** | **297** | **3** | **323** | **285** | **282** | **310** | **246** | **454** |

were generated from both *An. melas*/*An. gambiae* s.s.–*melas* hybrids and *An.* sp. X collected from Senguelen in the Maferinyah region. Analysis of *Wolbachia 16S rRNA* sequences obtained from *An. melas* and *An. gambiae* s.s.–*melas* hybrid individuals highlighted the occurrence of superinfections within this population, with the presence of multiple *Wolbachia* strains being indicated. The *Wolbachia 16S* sequences from some *An. melas* and hybrid individuals produced consensus sequences which were most closely related to *Wolbachia* strains of Supergroup A (such as *w*Mel, *w*AlbA and *w*Au) (therefore named *w*Anga-Guinea-A), of which two different A strains (named A1 and A2) could be determined in different individuals. By contrast, other *An. melas* and hybrid specimens produced *Wolbachia 16S* consensus sequences which grouped clearly with Supergroup B strains (*w*Anga-Guinea-B), also with two differing B strains able to be determined (B1 and B2) (figure 3). In addition, the sequence chromatograms from other *An. melas* and hybrid individuals consistently demonstrated mixed bases in the positions of variation between the *w*Anga-Guinea-A and *w*Anga-Guinea-B strains, with agreement both between forward and reverse sequence traces from the same individuals, as well as across multiple individuals, suggesting the presence of superinfections of both Supergroup A and B *Wolbachia* strains within these individuals. Repeat *Wolbachia 16S* sequencing for samples suggesting the presence of a *w*Anga-Guinea A and B strain superinfection did not allow confident separation of *An. melas* and hybrid individuals into *w*Anga-Guinea-A only, *w*Anga-Guinea-B only, or superinfected groups.

This, combined with the overall results from the *Wolbachia 16S* analysis from all infected individuals, suggested superinfections were widespread in *Wolbachia* positive individuals but there did not currently appear to be a clear dominant strain, or strain variant, which could be identified with the greater relative occurrence or apparent density (through consistent stronger sequencing signal strength) to the other strain(s) present in individuals from this population. This complexity was also mirrored when looking between *An. melas* and the hybrid specimens, with no clear distinction in the *Wolbachia* strain variants apparent in each group. Unfortunately, further comparative analysis of differing strains in *An. melas* and hybrid individuals using the *Wolbachia wsp* gene locus was not possible, as *wsp* sequence could only be successfully obtained from one *An. melas* individual, where *16S* analysis had indicated the presence of *w*Anga-Guinea-A1 only. This *wsp* sequence matched allele 23 within the *Wolbachia* MLST database (table 2), demonstrating that it is identical to the *wsp* sequences obtained from 20 other Supergroup A *Wolbachia* isolates contained within the database.

Phylogenetic analysis of both *Wolbachia 16S* and *wsp* gene fragments from *An.* sp. X indicated that the *w*AnsX strain is most closely related to *Wolbachia* strains of Supergroup B (such as *w*Pip, *w*AlbB, *w*AnsA, *w*AnM, *w*Ma and *w*No). Typing of the *w*AnsX *wsp* nucleotide sequence highlighted that there were no exact matches to *wsp* alleles currently in the *Wolbachia* MLST database (https://pubmlst.org/organisms/wolbachia-spp/), and only one of the four hypervariable regions (HVRs) matched a known sequence (HVR3: allele 3). All *Wolbachia* gene sequences of sufficient quality to generate a consensus were deposited into GenBank and accession numbers obtained (electronic supplementary material, tables S3 and S4).

*Wolbachia* MLST was undertaken to attempt to provide more accurate strain discrimination and phylogenies. This was successfully done for the novel *Anopheles Wolbachia* strains *w*Anga-Guinea-A1 and *w*AnsX although MLST gene fragment amplification was variable for *w*Anga-Guinea strains found in *An. melas* and *An. gambiae* s.s.–*melas* hybrids. The resultant MLST allelic profile for *w*Anga-Guinea-A1 (table 2) was closest to the profile for strain type 13, with the variation occurring in the two positions of mixed bases in the *hcpA* locus, being closest to *hcpA* allele 1, except for a change from G to A at position 313 and from A to G at position 319 on this locus (table 2). This may indicate the presence of both *hcpA* allele 1 and an *hcpA* variant. Even if *w*Anga-Guinea-A1 were identical to strain type 13, of the 19 records

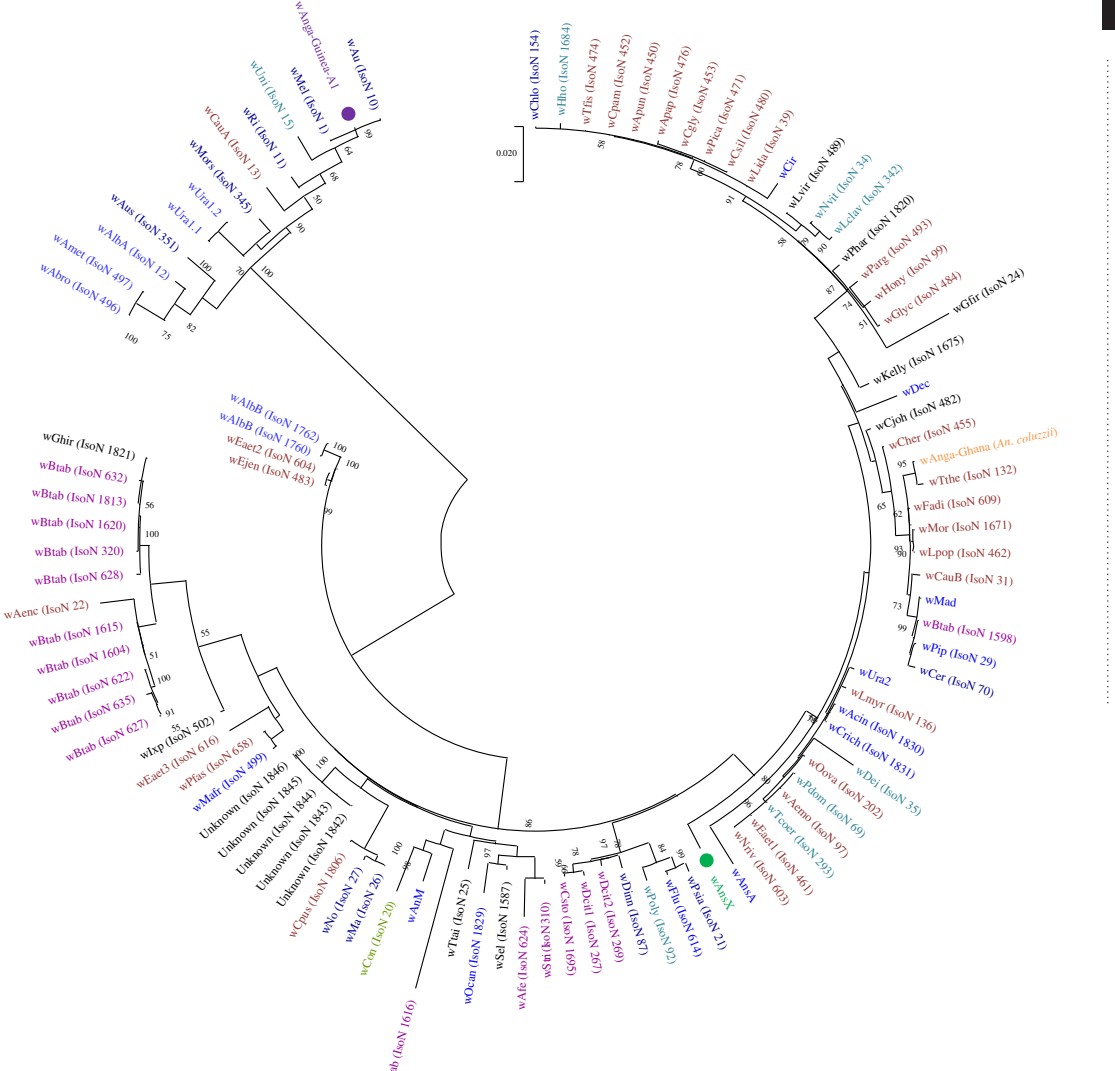

**Figure 4.** *Wolbachia* multilocus sequence typing (MLST) phylogenetic analysis of *w*Anga-Guinea-A1 and *w*AnsX. Maximum-likelihood molecular phylogenetic analysis from concatenation of all five MLST gene loci for resident *Wolbachia* strains *w*Anga-Guinea-A1 (in purple) and *w*AnsX (in green). Concatenated sequences obtained in this study are highlighted in bold with a filled circle node marker. The general time reversible model was used. A discrete Gamma distribution was used to model evolutionary rate differences among sites (5 categories (+G, parameter = 0.2484)). The rate variation model allowed for some sites to be evolutionarily invariable ([+I], 36.51% sites). The tree with the highest log likelihood (−10595.38) is shown and drawn to scale, with branch lengths measured in the number of substitutions per site. The analysis involved 102 nucleotide sequences. There was a total of 2079 positions in the final dataset. The concatenated MLST sequence data from *w*Anga-Ghana obtained from *An. coluzzii* in a previous study [26] is shown in orange. Concatenated sequence data from *Wolbachia* strains downloaded from the MLST database for comparison are shown with isolate numbers in brackets (IsoN). *Wolbachia* strains isolated from mosquito species are shown in blue, with those strains from other *Anopheles* species highlighted in bold. Strains isolated from other Dipteran species are shown in navy blue, from Coleoptera in olive green, from Hemiptera in purple, from Hymenoptera in teal blue, from Lepidoptera in maroon and from other, or unknown orders in black.

available on the MLST database (all Supergroup A), no other isolates with this strain type, where host information had been provided, were found in mosquito species. Concatenation of the MLST loci and phylogenetic analysis also confirms *w*Anga-Guinea-A1 is closest to strains belonging to Supergroup A, including *w*Mel and *w*AlbA (as also suggested by *16S* and *wsp* gene phylogenies). For *w*AnsX, new alleles for all five MLST gene loci (sequences differed from those currently present in the MLST database), and the therefore novel allelic profile, confirms this is a divergent novel *Wolbachia* strain (table 2). The phylogeny of *w*AnsX based on concatenated sequences of all five MLST gene loci confirms this strain clusters within Supergroup B and further demonstrates that it is distinct from other currently

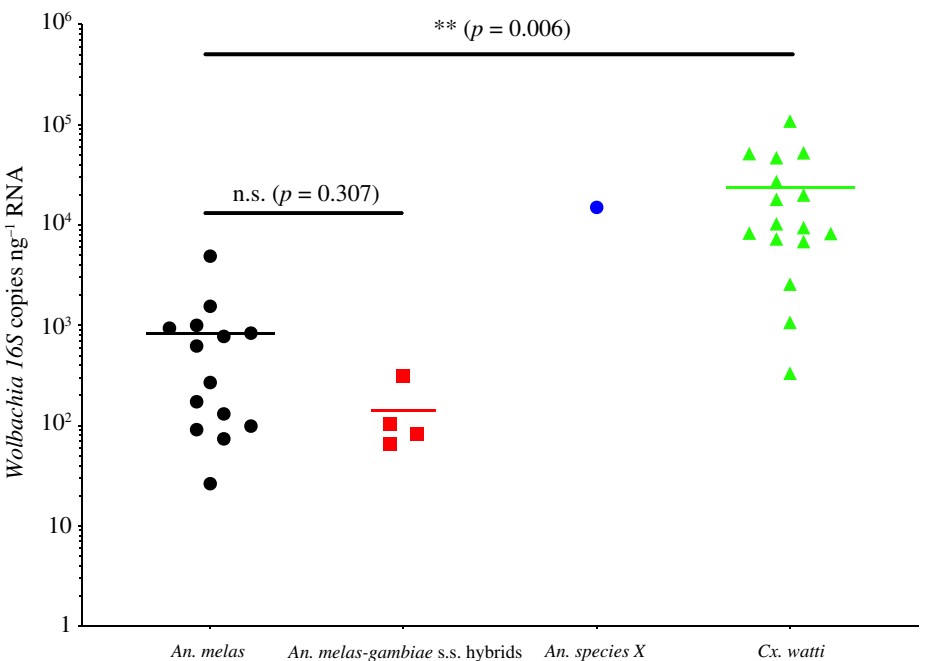

**Figure 5.** *Wolbachia* strain densities in wild-caught female mosquitoes from the Maferinyah sub-prefecture of Guinea. Total RNA extracted from individual mosquitoes was standardized to 2.0 ng μl$^{-1}$ prior to being used in qRT-PCR assays targeting the *Wolbachia 16S rRNA* gene. A synthetic oligonucleotide standard was designed to calculate *16S rRNA* gene copies per μl of RNA using a serial dilution series and all samples were run in triplicate in addition to no template controls.

available strain profiles (figure 4). Consistent with previous studies looking at novel *Wolbachia* strains in *Anopheles* species using MLST [24], these results highlight the lack of concordance between *Wolbachia* strain phylogeny and their insect hosts across diverse geographical regions.

### 3.3. *Wolbachia* strain densities and relative abundance

The relative densities of *Wolbachia* strains were estimated using qRT-PCR targeting the *16S rRNA* gene after first standardizing total RNA (ng per reaction). This allowed direct comparisons between phylogenetically diverse *Anopheles* species and accounts for variation in mosquito body size and RNA extraction efficiency between samples. This also allows a comparison with another novel natural *Wolbachia* strain present in *Cx. watti* (termed *w*Wat strain) collected in Maferinyah, contemporaneously with the *Anopheles* specimens. *16S rRNA* qRT-PCR analysis revealed a mean of $1.50 \times 10^4$ ($\pm 4.37 \times 10^3$) *16S rRNA* copies μl$^{-1}$ for the *w*AnsX strain in the single individual (figure 5, electronic supplementary material, electronic supplementary material table S5). Lower mean densities were found for the *w*Anga-Guinea strains in *An. melas* individuals ($n = 14$) and *An. gambiae* s.s.–*melas* hybrids ($n = 4$) with $8.20 \times 10^2$ ($\pm 2.90 \times 10^2$) and $1.41 \times 10^2$ ($\pm 3.95 \times 10^1$) *16S rRNA* copies μl$^{-1}$, respectively. The densities were compared with the *w*Wat strain in *Cx. watti* females also collected in the Maferinyah region with a mean density of $2.37 \times 10^4$ ($\pm 5.99 \times 10^3$). The density of the *w*Wat strain was significantly higher than the *w*Anga-Guinea strains found in *An. melas* and hybrids (unpaired *t*-test, $p = 0.002$). Individual *An. gambiae* s.s. extracts from the Faranah region that were identified as *Wolbachia*-infected by amplification of the *16S rRNA* gene [47] did not result in any *16S rRNA* qRT-PCR amplification, suggesting a very low density *Wolbachia* strain present in these individuals.

### 3.4. *Wolbachia* and *Asaia* co-infections

Individual mosquitoes shown to be infected with the *w*AnsX or *w*Anga-Guinea strains were screened for the presence of *Asaia* bacteria using qRT-PCR. Co-infections were detected in all *An. melas* ($n = 14$, mean *Asaia 16S rRNA* Ct value = 30.60 ± 2.02), all *An. gambiae* s.s.–*melas* hybrids ($n = 4$, mean *Asaia 16S rRNA* Ct value = 26.32 ± 3.54) and in the single *An. species X* (*Asaia 16S rRNA* Ct value = 34.92) (electronic supplementary material, table S5).

# 4. Discussion

Endosymbiotic *Wolbachia* bacteria are particularly widespread through insect populations but were historically considered absent from the *Anopheles* genera [19]. The discovery of additional novel natural strains of *Wolbachia* in *Anopheles* species suggests that the prevalence and diversity has been significantly under-reported to date. Since 2014, there have been several reports of detection of *Wolbachia* strains in major malaria vectors, such as sibling species in the *An. gambiae* complex [25,26,28–30] and *An. moucheti* [26]. This study provides evidence for *Wolbachia* strains in *An. melas*, a species within the *An. gambiae* complex, which can be an important local vector of malaria in West-African coastal areas where it breeds in brackish water, mangrove forests and salt marshes [59,60]. Its importance as a local malaria vector was shown in Equatorial Guinea where the average number of malaria infective *An. melas* bites/person/year was recorded at up to 130 [61]. The finding of natural *An. gambiae* s.s.–*melas* hybrids in this study appears highly unusual, with published accounts of hybridization between members of the *An. gambiae* complex seeming to agree that detection of hybrids in wild populations is relatively rare [62], and when it does occur, seems most often to be a combination of hybrids between *An. gambiae* s.s., *An. coluzzii* or *An. arabiensis*. Historical reports of *An. gambiae* s.s.–*melas* hybrids were also in West Africa but with laboratory colonies, giving variable results for ongoing success of hybrid colonies [63–65]. Interestingly colonized *An. melas* and F1 hybrid larvae were able to be reared in distilled water in the laboratory, rather than requiring a higher salinity content as might be expected from the natural ecology of *An. melas* [65]. As *An. melas* is more geographically restrained and has a more defined ecological niche than other members of the *An. gambiae* complex, natural hybrids composed of these constituent species are arguably less likely to occur, with fewer areas of sympatry. Natural hybrids may also be underestimated [66] due to sampling bias with a greater proportion of studies focusing on the more widely distributed major anthrophilic malaria vectors, *An. gambiae* and *An. arabiensis* [67].

Hybrid detection is also dependent on the methodology used for species identification and the format of species-specific diagnostic assays [40,62]. Our testing highlighted that amplification and clarity of hybrid detection was improved with the use of the ribosomal IGS PCR primers [40] for each species in single-plex format, rather than the standard higher throughput multiplex format, where primers for multiple members of the *An. gambiae* complex are included at the same time, with different product sizes for species discrimination. This is unsurprising due to the designed aims of the multiplex assay, and potential variations in reaction efficiency between species, particularly when hybridized, which were highlighted in the original publication [40]. However, this could potentially result in reduced detection of natural hybrids, compared with the apparent detection of individual species, when used for widespread screening and species identification. Sanger sequencing of the single-plex species-specific IGS PCR products for representative hybrid samples enabled confirmation of the hybridization and the avoidance of doubt from any possibility of specificity problems [40,62], before further confirmation was obtained through subsequent sequencing and phylogenetic analysis of other gene fragments.

Genetic divergence also probably affects interspecific hybrids, and the original delineation of the member species within the *An. gambiae* complex was concluded on the basis of hybrid male sterility from early crossing experiments [67]. However, the full extent and impacts of interspecific hybridization between members of the complex is still under investigation and debate [66]. *Anopheles gambiae* s.s. and *An. melas* have a greater degree of genetic divergence from one another when compared with other members of the complex (such as *An. gambiae* s.s., *An. coluzzii* and *An. arabiensis)* and *An. melas* groups separately and more closely to *An. merus* and *An. quadriannulatus* sequences. Even within *An. melas*, species-specific microsatellite markers and mitochondrial genetic analysis of geographically distinct populations suggested there was species-level divergence between different populations, resulting in three distinct major clusters; Bioko Island, Western mainland and Southern mainland African populations (with mainland population division occurring in Cameroon) [67]. In the context of the results of this study, our *An. melas* would be included in the Western mainland cluster (this is supported by our phylogenetic analysis). Following the discovery of *Wolbachia* in this population, it would be interesting to investigate whether *Wolbachia* strains were also present in other *An. melas* geographic clusters, and whether the CI phenotype was evident in some or all of these strains. If stable *Wolbachia* infections were present in some populations but not others, it also raises the question of the length of time *Wolbachia* may have been present in this species and whether *Wolbachia* infections may be having an influence on the host population genetics and affecting genetic divergence and speciation over time.

This study and previous studies have highlighted how thorough and accurate molecular identification of mosquito specimens is important given the difficulties of morphological identification, the potential for currently unrecognized cryptic species [58,68] and potential for inaccuracies for certain species where only diagnostic species PCR-based methods are used for molecular identification [69]. The discovery of the *w*AnsX strain through PCR screening and Sanger sequencing led to the retrospective confirmation of the host mosquito species using Sanger sequencing as all individuals that were *Wolbachia* positive by PCR were initially morphologically identified as members of the *An. gambiae* complex. Phylogenetic analysis and confident species discrimination is dependent on the sequences available for comparison at the time. Sequencing and phylogenetic analysis of all three regions for this specimen indicated placement within the *Cellia* subgenus and Myzomyia Series of *Anopheles*, with the greatest number of closely related comparative sequences available for comparison in the ITS2 region. Our analysis revealed that this species is closest to *Anopheles* sp. 7, followed by *An. theileri* from sequences currently available. *Anopheles* sp. 7 BSL-2014 was collected in the western Kenyan Highlands, with 1 of 23 specimens *P. falciparum* ELISA sporozoite and PCR positive [58]. *Anopheles theileri* was collected in the Democratic Republic of Congo [70] and was found to be infected with *Plasmodium* sporozoites in eastern Zambia [71].

The results of this study also highlight the requirement to provide as much genetic information and confirmation as possible for a newly discovered strain of *Wolbachia* (particularly low-density infections). The first discovery of *Wolbachia* strains in wild *An. gambiae* populations in Burkina Faso resulted from sequencing of the *16S rRNA* gene rather than screening using *Wolbachia*-specific genes [25]. A more recent comprehensive analysis through screening of *An. gambiae* genomes (Ag1000G project) concluded that determining whether a *Wolbachia* strain is present in a given host based on the sequencing of one gene fragment (often *16S rRNA*) is problematic and caution should be taken [31]. In this study, from *An. gambiae* s.s., we were only able to amplify a *Wolbachia 16S rRNA* gene fragment, which is consistent with numerous recent studies in which low-density strains have been detected [27,30]. As a result, caution must be taken in drawing conclusions on the stability of infection and biological significance. Other explanations for the amplification of *16S rRNA* gene fragments include *Wolbachia* DNA insertions into an insect chromosome or contamination from non-mosquito material such as ectoparasites or plants [31]. In contrast with previous studies, we extracted RNA, increasing the chances that detection of the *16S rRNA* gene is from actively expressed *Wolbachia* and indicating amplification is more likely of bacterial gene origin (rather than through integration into the host genome). Although RNA extraction kits are optimized for RNA and we measured high levels of total RNA using a fluorometer, there is a small possibility that our amplification and sequencing could result from co-extracted *Wolbachia* gDNA. Detection and sequencing of *Wolbachia* gDNA have been used previously in numerous studies to characterize strain phylogenies [25–29]. However, to provide greater confidence in the expression of *Wolbachia* genes in future work, a DNase treatment could be undertaken to ensure amplification is only resulting from the cDNA. Regardless, these results are consistent with previous studies in which every *Wolbachia 16S rRNA* amplicon and sequence attributed to *An. gambiae* s.s. is unique and appears at very low density [31]. Therefore, further experiments are needed to confirm these strains are genuine endosymbionts in their hosts such as microscopy and genome sequencing [33].

The densities of the *w*Anga-Guinea and *w*AnsX strains detected in Senguelen (measured using qRT-PCR) are significantly higher than *Wolbachia* detected in *An. gambiae* s.s. from Faranah (which were not detectable using this qRT-PCR assay targeting the *16S rRNA* gene). The *w*Anga-Guinea strains appear to have both an intermediate prevalence rate and density and further studies are required to elucidate the relative density contribution and possible differential localization of these *Wolbachia* strains within the mosquito host, whether these strains may be influencing host population genetics (including the occurrence of natural hybrids and the intraspecific diversity within *An. melas*) and investigate these strains across more diverse geographical areas. Unfortunately, more extensive separate characterization and determination of relative densities of the *Wolbachia* strains within superinfected *An. melas* and *An. gambiae* s.s.–*melas* hybrids was not possible in this study. Further work, through the design of strain-specific PCRs and cloning followed by sequencing of genes from the separate strains, would help to expand knowledge on the characteristics and possibilities for the further potential use of these strains. Caution and further investigation is also required for the *w*AnsX strain, as this was detected from the only collected individual of this unclassified *Anopheles* species. The detection of *Wolbachia*-*Asaia* co-infections in all individuals was in contrast with our previous study [26] but *Asaia* can be environmentally acquired at different mosquito life stages and the prevalence and density were significantly variable across different *Anopheles* species and locations [26]. These contrasting results suggest a complex association between these two bacterial species in wild *Anopheles* mosquito populations, and given that *Asaia* is environmentally acquired, this association will be highly location dependent.

*Wolbachia* strains in *An*. species A (*w*AnsA) and *An. moucheti* (*w*AnM) [26], and now *An. melas* (*w*Anga-Guinea-A1) and *An.* sp. X (*w*AnsX), have complete MLST and *wsp* profiles and are at significantly higher densities when compared with strains detected in *An. gambiae* s.s. from the same countries. As *Wolbachia* density is strongly correlated with arbovirus inhibition in *Aedes* mosquitoes [5,7,11,12], higher density strains in *Anopheles* species would be predicted to have a greater impact on malaria transmission in field populations. In this study, we screened for *P. falciparum* infection and found very low prevalence rates (less than 1%; data not shown) preventing any statistical analysis on *Wolbachia–Plasmodium* interactions. This study and previous studies measuring a direct impact on *Plasmodium* infection in wild populations are dependent on parasite infection rates which can be low even in malaria-endemic areas [26] and particularly for the infective sporozoite stage [72]. Low pathogen prevalence rates are also limiting factors in assessing the effect of natural strains of *Wolbachia* on arboviruses in wild mosquito populations [73]. In addition to looking at effects on *Plasmodium* prevalence in field populations, further work should look to undertake vector competence experiments with colonized populations and to determine if these *Wolbachia* strains are present in tissues such as the midgut and salivary glands which are critical to sporogony. Further studies are also needed to determine if the *w*Anga-Guinea strains are maternally transmitted given our results would suggest they are likely to be from the *An. melas*, rather than from *An. gambiae* s.s. Furthermore, an assessment of how these *Wolbachia* strains are being maintained in field populations is needed, and to determine if the CI reproductive phenotype can be induced by these strains (and if it affects the viability of subsequent generations). As the chances of success of *Wolbachia* transinfection experiments can be improved by the adaptation of the *Wolbachia* strain to the target host genetic background [74], this may imply a favourable potential for *w*Anga-Guinea transinfection experiments and successful establishment of a stable *Wolbachia* infection within *An. gambiae* s.s. colonies, in addition to *An. melas*. If achievable, this would be a big step forward in determining whether these strains (which appear relatively higher density than *Wolbachia* previously detected in the *An. gambiae* complex) could reduce malaria transmission through *Wolbachia*-based biocontrol strategies.

# 5. Conclusion

Although the debate continues over the biological significance (or even presence) of natural strains in the *An. gambiae* complex, this study provides strong evidence of additional novel strains with relatively higher density infections, in addition to *Wolbachia* positive natural hybrids in the *An. gambiae* complex, and may reflect the under-reporting of natural strains in the *Anopheles* genus. The presence of *Wolbachia* superinfections increases the complexity of phylogenetic characterization of individual strains and the determination of the relative contribution of each strain to the overall density. There are previous studies showing natural *Wolbachia* superinfections in wild mosquito populations such as *Ae. albopictus* [75] and superinfections have been generated in mosquitoes used for biocontrol strategies [7,76], indicating superinfections can form stable associations with mosquito hosts. Candidate *Wolbachia* strains for mosquito biocontrol strategies require synergistic phenotypic effects to impact the transmission of mosquito-borne pathogens and further studies are needed to determine if these strains would induce CI and what effects they may have on host fitness. Whether these *Wolbachia* superinfections can inhibit *Plasmodium* parasites [28,29] or influence the ability to transinfect other *Wolbachia* strains for population suppression and replacement strategies [77] remains to be determined, but further investigation is warranted.

Ethics. Mosquito collection protocols were reviewed and approved by the Comite National d'Ethique pour la Recherche en Sante (030/CNERS/17) and the institutional review boards (IRB) of the London School of Hygiene and Tropical Medicine (nos. 14798 and 15127) and the Centers for Disease Control and Prevention, USA (2018-086); all study procedures were performed in accordance with relevant guidelines and regulations. Fieldworkers participating in human landing catches were provided with malaria prophylaxis for the duration of the study.

Data accessibility. The datasets supporting this article have been uploaded as part of the electronic supplementary material and all the sequencing data generated is available in GenBank with accession numbers as shown in the relevant electronic supplementary material, tables.

Authors' contributions. C.L.J. designed the study, carried out the analyses and drafted the manuscript. C.C.-U. performed fieldwork, carried out preliminary laboratory analysis and reviewed the manuscript. A.H.B. supervised fieldwork and reviewed the manuscript. C.S. performed fieldwork and reviewed the manuscript. E.K.L. supervised fieldwork and reviewed the manuscript. M.K. performed fieldwork and reviewed the manuscript. S.R.I. supervised fieldwork and reviewed the manuscript. T.W. designed the study, carried out analyses, drafted the manuscript and provided overall supervision.

Competing interests. We declare we have no competing interests.

Funding. C.L.J. and T.W. were supported by Wellcome Trust /Royal Society grants awarded to T.W. (101285/Z/13/Z): http://www.wellcome.ac.uk; https://royalsociety.org. S.R.I. was supported by the President's Malaria Initiative (PMI)/CDC.

Acknowledgements. The following people are thanked for their efforts in the original collection of mosquitoes: Moussa Sylla, Gnepou Camara, Louisa Messenger, Patrick Heard, Yaya Barry, Denka Camara and Ismael Yansane. This publication made use of the PubMLST website (https://pubmlst.org/Wolbachia/) sited at the University of Oxford (Jolley & Maiden 2010, BMC Bioinformatics, 11:595). The development and maintenance of this site has been funded by the Wellcome Trust.

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
