## [Peer Review File · Royal Society Open Science]

Review History

RSOS-202032.R0 (Original submission)

Review form: Reviewer 1 (Michael Gerth)

Is the manuscript scientifically sound in its present form?

Yes

Are the interpretations and conclusions justified by the results?

Yes

Is the language acceptable?

Yes

Do you have any ethical concerns with this paper?

No

Have you any concerns about statistical analyses in this paper?

No

Recommendation?

Major revision is needed (please make suggestions in comments)

Comments to the Author(s)

This is a relatively straightforward molecular screen for Wolbachia symbionts in a collection of Anopheles mosquitoes. The main findings are that some specimens were PCR positive for Wolbachia, and that there is molecular evidence for hybridization between Anopheles melas and An. gambiae ss. The study is one of several similar recently published articles and is plagued by the same problem: the evidence for Wolbachia is entirely based on PCR, and this comes with many problems which were discussed in detail here: <https://doi.org/10.1128/mBio.00784-19>. Similar to previous studies, sequencing of Wolbachia loci was unsuccessful in some cases and titres of Wolbachia DNA were low; so overall the biological relevance of some of the findings is unclear. The high titre Wolbachia infections in An. melas and sp. X do appear genuine given the data. Further novel, and potentially more interesting aspects are the detection of potential hybridization events between two Anopheles species, and the molecular detection of a potentially novel Anopheles species. I think that the paper can be improved in several points, and I would like to ask the authors to consider two main issues:

- 1) More cautious interpretation of PCR results: I think it is misleading to claim Wolbachia detection if sequencing was unsuccessful. Samples that are PCR positive, but failed to produce a sequence should be conservatively interpreted as negative. In the case of the An. gambiae s.s samples, there was no sequence, and no amplification in the qPCR experiments. Nevertheless, Wolbachia strains in An. gambiae ss are highlighted in the abstract, and this should be removed. Using RNA extracts instead of DNA may contribute to reducing the number of false Wolbachia positives. However, this would only be efficient with DNase treatment after RNA extractions to remove residual DNA (which is always present irrespective of the RNA extraction protocol). I couldn't find any information on this in the manuscript, so it would be good to see clarification here. Overall, the limitations should be highlighted clearly and avenues discussed for confirming the potential Wolbachia infections.
- 2) More concise presentation: The paper is quite long with many figures, and I believe that a more concise presentation would be beneficial. For example, is it really necessary to have separate tree figures in the main manuscript for each mitochondrial locus and the nuclear loci plus a tree from a concatenated dataset? All mitochondrial genes are essentially the same locus, and the trees don't differ a lot anyway. I would recommend to show a single tree here, and move the other figures into the supplement. The same is true for the Wolbachia loci: why so many different trees when the main goal is supergroup assignment and for that a single tree would suffice. I also recommend to remove the wsp tree entirely, as this locus is useless as phylogenetic marker. In the results section, there lengthy paragraphs about technical details that overall don't contribute much to the overall conclusions, and my suggestion would be to focus on the main findings to improve accessibility for readers. More details on this below.

Minor comments, hopefully constructive:

P3, L24-28

While the limitations of these studies are mentioned in the discussion, I feel it would make sense to highlight them in the introduction as well.

P3, L59-60

Did this protocol include DNase treatment?

P4, L5-6

Did you use a Ct threshold after which detection was considered unreliable?

P4, L44-45

This reads as if the 16S PCR was done on genomic DNA rather than cDNA. Please clarify.

P4, L46-47

The CifA/CifB PCRs are not mentioned anywhere in the results or discussion. Suggest to remove the protocol here.

P5, L38

Please provide justification for using TN model of evolution.

P5, L39-40

If this means bootstrap analysis, please provide details here.

P5, L41-44

It appears the description is for the calculation of the starting tree, rather than for the ML reconstructions. Please clarify.

P5, L49ff

Please provide more details on negative and positive controls here. Did you include Wolbachia-free Anopheles specimens as negative controls?

P7, L18

It appears there are 2 novel supergroup A strains and 2 belonging to supergroup B. The accession numbers for the B strains seem to be missing from Table S3

P7, L26-50

Is this level of detail really necessary? I think it is almost impossible to infer the relative abundances of two taxa in a sanger chromatogram from a mixed sample. I would suggest to either perform qPCRs for the different strains or to omit these paragraphs.

P8, L7-21

Again, I think these technical details are not very important. Suggest to consider removing this paragraph.

P8, L32

Suggest to change "diverse" to "divergent".

P9, L37-38

Please explain why singleplex approaches would lead to reduced sensitivity for the detection of hybrids, this is not immediately apparent

P10, L6-7

Unclear what is meant by this statement

P10, L7-10

Sentence appears incomplete. Suggest to rephrase.

P10, L21-22

Or rather, the results highlight the necessity for other types of evidence in addition to the molecular data

Review form: Reviewer 2

Is the manuscript scientifically sound in its present form?

Yes

Are the interpretations and conclusions justified by the results?

Yes

Is the language acceptable?

Yes

Do you have any ethical concerns with this paper?

No

Have you any concerns about statistical analyses in this paper?

No

Recommendation?

Accept with minor revision (please list in comments)

Comments to the Author(s)

Page 3, Line 32: "Interestingly, the presence of Wolbachia strains in Anopheles was inversely correlated to other bacteria species such as Asaia that are stably associated with several species". This work <https://www.nature.com/articles/s41598-020-70745-0> also shed light on Wolbachia-microbiome associations and could be added to the other works cited here.

Page 5, Line 58: It is unclear how much cDNA was used for qPCR. In the methods it is indicated 2uL, please specify the amount in ng not volume. In the results, it is stated this was normalized to ng per reaction. This should be consistent with the methods described to improve reproducibility of this study.

Wolbachia superinfection: the author have come to this conclusion by observing mixed bases in the sequenced but could not confirm the dominant strain. It is unclear why the authors have not cloned the gene/s from the samples and sequenced multiple clones to resolve which strain is the dominant one. If this is not possible for the authors to conduct this analysis now, I suggest that this approach is performed in the future.

Decision letter (RSOS-202032.R0)

Dear Dr Walker

The Editors assigned to your paper RSOS-202032 "Evidence for natural hybridisation and novel Wolbachia strain superinfections in the *Anopheles gambiae* complex from Guinea" have now received comments from reviewers and would like you to revise the paper in accordance with the

reviewer comments and any comments from the Editors. Please note this decision does not guarantee eventual acceptance.

Please submit your revised manuscript and required files (see below) no later than 21 days from today's (ie 25-Jan-2021) date. Note: the ScholarOne system will 'lock' if submission of the revision is attempted 21 or more days after the deadline. If you do not think you will be able to meet this deadline please contact the editorial office immediately.

on behalf of Dr Krijn Paaijmans (Associate Editor) and Kevin Padian (Subject Editor)
openscience@royalsociety.org

Reviewer comments to Author:

Reviewer: 1
Comments to the Author(s)

This is a relatively straightforward molecular screen for Wolbachia symbionts in a collection of Anopheles mosquitoes. The main findings are that some specimens were PCR positive for Wolbachia, and that there is molecular evidence for hybridization between *Anopheles melas* and *An. gambiae* ss. The study is one of several similar recently published articles and is plagued by the same problem: the evidence for Wolbachia is entirely based on PCR, and this comes with many problems which were discussed in detail here: <https://doi.org/10.1128/mBio.00784-19>. Similar to previous studies, sequencing of Wolbachia loci was unsuccessful in some cases and titres of Wolbachia DNA were low; so overall the biological relevance of some of the findings is unclear. The high titre Wolbachia infections in *An. melas* and sp. X do appear genuine given the data. Further novel, and potentially more interesting aspects are the detection of potential hybridization events between two *Anopheles* species, and the molecular detection of a potentially

novel *Anopheles* species. I think that the paper can be improved in several points, and I would like to ask the authors to consider two main issues:

1) More cautious interpretation of PCR results: I think it is misleading to claim *Wolbachia* detection if sequencing was unsuccessful. Samples that are PCR positive, but failed to produce a sequence should be conservatively interpreted as negative. In the case of the *An. gambiae* s.s samples, there was no sequence, and no amplification in the qPCR experiments. Nevertheless, *Wolbachia* strains in *An. gambiae* ss are highlighted in the abstract, and this should be removed. Using RNA extracts instead of DNA may contribute to reducing the number of false *Wolbachia* positives. However, this would only be efficient with DNase treatment after RNA extractions to remove residual DNA (which is always present irrespective of the RNA extraction protocol). I couldn't find any information on this in the manuscript, so it would be good to see clarification here. Overall, the limitations should be highlighted clearly and avenues discussed for confirming the potential *Wolbachia* infections.

2) More concise presentation: The paper is quite long with many figures, and I believe that a more concise presentation would be beneficial. For example, is it really necessary to have separate tree figures in the main manuscript for each mitochondrial locus and the nuclear loci plus a tree from a concatenated dataset? All mitochondrial genes are essentially the same locus, and the trees don't differ a lot anyway. I would recommend to show a single tree here, and move the other figures into the supplement. The same is true for the *Wolbachia* loci: why so many different trees when the main goal is supergroup assignment and for that a single tree would suffice. I also recommend to remove the *wsp* tree entirely, as this locus is useless as phylogenetic marker. In the results section, there lengthy paragraphs about technical details that overall don't contribute much to the overall conclusions, and my suggestion would be to focus on the main findings to improve accessibility for readers. More details on this below.

Minor comments, hopefully constructive:

P3, L24–28

While the limitations of these studies are mentioned in the discussion, I feel it would make sense to highlight them in the introduction as well.

P3, L59–60

Did this protocol include DNase treatment?

P4, L5–6

Did you use a Ct threshold after which detection was considered unreliable?

P4, L44–45

This reads as if the 16S PCR was done on genomic DNA rather than cDNA. Please clarify.

P4, L46–47

The *CifA*/*CifB* PCRs are not mentioned anywhere in the results or discussion. Suggest to remove the protocol here.

P5, L38

Please provide justification for using TN model of evolution.

P5, L39–40

If this means bootstrap analysis, please provide details here.

P5, L41–44

It appears the description is for the calculation of the starting tree, rather than for the ML reconstructions. Please clarify.

P5, L49ff

Please provide more details on negative and positive controls here. Did you include Wolbachia-free Anopheles specimens as negative controls?

P7, L18

It appears there are 2 novel supergroup A strains and 2 belonging to supergroup B. The accession numbers for the B strains seem to be missing from Table S3

P7, L26-50

Is this level of detail really necessary? I think it is almost impossible to infer the relative abundances of two taxa in a sanger chromatogram from a mixed sample. I would suggest to either perform qPCRs for the different strains or to omit these paragraphs.

P8, L7-21

Again, I think these technical details are not very important. Suggest to consider removing this paragraph.

P8, L32

Suggest to change "diverse" to "divergent".

P9, L37-38

Please explain why singleplex approaches would lead to reduced sensitivity for the detection of hybrids, this is not immediately apparent

P10, L6-7

Unclear what is meant by this statement

P10, L7-10

Sentence appears incomplete. Suggest to rephrase.

P10, L21-22

Or rather, the results highlight the necessity for other types of evidence in addition to the molecular data

Reviewer: 2

Comments to the Author(s)

Page 3, Line 32: "Interestingly, the presence of Wolbachia strains in Anopheles was inversely correlated to other bacteria species such as Asaia that are stably associated with several species". This work <https://www.nature.com/articles/s41598-020-70745-0> also shed light on Wolbachia-microbiome associations and could be added to the other works cited here.

Page 5, Line 58: It is unclear how much cDNA was used for qPCR. In the methods it is indicated 2uL, please specify the amount in ng not volume. In the results, it is stated this was normalized to ng per reaction. This should be consistent with the methods described to improve reproducibility of this study.

Wolbachia superinfection: the author have come to this conclusion by observing mixed bases in the sequenced but could not confirm the dominant strain. It is unclear why the authors have not

cloned the gene/s from the samples and sequenced multiple clones to resolve which strain is the dominant one. If this is not possible for the authors to conduct this analysis now, I suggest that this approach is performed in the future.

===PREPARING YOUR MANUSCRIPT===

===PREPARING YOUR REVISION IN SCHOLARONE===

<https://royalsociety.org/journals/authors/author-guidelines/#supplementary-material> to include a suitable title and informative caption. An example of appropriate titling and captioning may be found at https://figshare.com/articles/Table_S2_from_Is_there_a_trade-off_between_peak_performance_and_performance_breadth_across_temperatures_for_aerobic_sc_ope_in_teleost_fishes_/3843624.

Author's Response to Decision Letter for (RSOS-202032.R0)

See Appendices A & B.

Decision letter (RSOS-202032.R1)

Dear Dr Walker,

It is a pleasure to accept your manuscript entitled "Evidence for natural hybridisation and novel Wolbachia strain superinfections in the *Anopheles gambiae* complex from Guinea" in its current form for publication in Royal Society Open Science.

on behalf of Dr Krijn Paaijmans (Associate Editor) and Kevin Padian (Subject Editor)
openscience@royalsociety.org

Associate Editor Comments to Author (Dr Krijn Paaijmans):
Associate Editor

Comments to the Author:

The authors did a great job addressing the comments of both reviewers. The major issues have been adequately addressed: (1) a more cautious interpretation of PCR results, (2) the need for a DNase treatment in future experiments to ensure amplification is resulting from only cDNA, and (3) a modification of the manuscript to make it more concise.

Appendix A

We would like to thank the reviewer for a very constructive and useful review that has improved our manuscript. Our responses to reviewer 2 are shown as follows in **bold** and changes to the manuscript are highlighted in yellow:

Reviewer: 2

Comments to the Author(s)

Page 3, Line 32: “Interestingly, the presence of Wolbachia strains in Anopheles was inversely correlated to other bacteria species such as Asaia that are stably associated with several species”. This work <https://www.nature.com/articles/s41598-020-70745-0> also shed light on Wolbachia-microbiome associations and could be added to the other works cited here.

Thank you for this suggestion and we have added this reference.

Page 5, Line 58: It is unclear how much cDNA was used for qPCR. In the methods it is indicated 2uL, please specify the amount in ng not volume. In the results, it is stated this was normalized to ng per reaction. This should be consistent with the methods described to improve reproducibility of this study.

Apologies and thank you for spotting this mistake. As we used ‘RNA’ in our onestep qRT-PCR assays, we have corrected this error (replaced ‘DNA’ with ‘RNA’). Earlier in this paragraph we have the details that we used 2.0 ng/uL RNA in each reaction.

Wolbachia superinfection: the author have come to this conclusion by observing mixed bases in the sequenced but could not confirm the dominant strain. It is unclear why the authors have not cloned the gene/s from the samples and sequenced multiple clones to resolve which strain is the dominant one. If this is not possible for the authors to conduct this analysis now, I suggest that this approach is performed in the future

Thank you for this useful suggestion, unfortunately it wasn’t possible to attempt this but we agree this would be very useful for further work and therefore have included a sentence within the discussion regarding further work for greater characterisation of the superinfection strains.

Appendix B

We would like to thank the reviewer for a very constructive and useful review that has improved our manuscript. Our responses to reviewer 1 are shown as follows in **bold** and changes to the manuscript are highlighted in yellow:

Reviewer: 1

Comments to the Author(s)

This is a relatively straightforward molecular screen for *Wolbachia* symbionts in a collection of *Anopheles* mosquitoes. The main findings are that some specimens were PCR positive for *Wolbachia*, and that there is molecular evidence for hybridization between *Anopheles melas* and *An. gambiae* ss. The study is one of several similar recently published articles and is plagued by the same problem: the evidence for *Wolbachia* is entirely based on PCR, and this comes with many problems which were discussed in detail here: <https://doi.org/10.1128/mBio.00784-19>. Similar to previous studies, sequencing of *Wolbachia* loci was unsuccessful in some cases and titres of *Wolbachia* DNA were low; so overall the biological relevance of some of the findings is unclear. The high titre *Wolbachia* infections in *An. melas* and sp. X do appear genuine given the data. Further novel, and potentially more interesting aspects are the detection of potential hybridization events between two *Anopheles* species, and the molecular detection of a potentially novel *Anopheles* species. I think that the paper can be improved in several points, and I would like to ask the authors to consider two main issues:

1) More cautious interpretation of PCR results: I think it is misleading to claim *Wolbachia* detection if sequencing was unsuccessful. Samples that are PCR positive, but failed to produce a sequence should be conservatively interpreted as negative. In the case of the *An. gambiae* s.s samples, there was no sequence, and no amplification in the qPCR experiments. Nevertheless, *Wolbachia* strains in *An. gambiae* ss are highlighted in the abstract, and this should be removed.

We agree that for *An. gambiae* ss without amplification in qPCR experiments and failed sequencing attempts we should be more conservative and interpret as negative. We have adjusted our summary (abstract) accordingly as we agree that detection of *Wolbachia* DNA sequences from these samples should not be presented in the abstract as a major result of this manuscript.

Using RNA extracts instead of DNA may contribute to reducing the number of false *Wolbachia* positives. However, this would only be efficient with DNase treatment after RNA extractions to remove residual DNA (which is always present irrespective of the RNA extraction protocol). I couldn't find any information on this in the manuscript, so it would be good to see clarification here. Overall, the limitations should be highlighted clearly and avenues discussed for confirming the potential *Wolbachia* infections.

Although we did not include a DNase treatment the Qiagen RNeasy 96 kit is optimised for RNA extraction. However, we completely agree that this does not fully prevent the possibility of dsDNA from being present and amplified in our one step RT PCR assays (in which a normalised RNA amount is added to the reaction). We have modified our existing discussion on this for clarity by adding the following sentences:

“Although RNA extraction kits are optimised for RNA and we measured high levels of total RNA using a Fluorometer, there is a small possibility that our amplification and sequencing could result from co-extracted *Wolbachia* gDNA. Detection and sequencing of *Wolbachia* gDNA has been used previously in numerous studies to characterise strain phylogenies [25-29]. However, to provide greater confidence on expression of *Wolbachia* genes in future work, a DNase treatment could be undertaken to ensure amplification is resulting from only cDNA.”

2) More concise presentation: The paper is quite long with many figures, and I believe that a more concise presentation would be beneficial. For example, is it really necessary to have separate tree figures in the main manuscript for each mitochondrial locus and the nuclear loci plus a tree from a concatenated dataset? All mitochondrial genes are essentially the same locus, and the trees don't differ a lot anyway. I would recommend to show a single tree here, and move the other figures into the supplement. The same is true for the *Wolbachia* loci: why so many different trees when the main goal is supergroup assignment and for that a single tree would suffice. I also recommend to remove the *wsp* tree entirely, as this locus is useless as phylogenetic marker. In the results section, there lengthy paragraphs about technical details that overall don't contribute much to the overall conclusions, and my suggestion would be to focus on the main findings to improve accessibility for readers. More details on this below.

We have modified our manuscript to be much more concise as follows:

- 1. We now have 5 main figures and have moved 3 phylogenetic trees to supplementary figures (COI/COII, ND4-ND5 and *wsp*). Although we do agree that the recombination rate of *wsp* does mean for phylogenetic purposes it's not a great marker, it does show that these new strains are novel and we are not detecting previously seen strains (so we feel warrants inclusion in the supplementary information).**
- 2. We have significantly reduced the technical details in the results as we agree that this improves the accessibility for readers (more specific details are shown further in responses to minor comments).**

Minor comments, hopefully constructive:

P3, L24–28

While the limitations of these studies are mentioned in the discussion, I feel it would make sense to highlight them in the introduction as well.

We have added some sentences to highlight the limitations of these studies:

“As the majority of studies have used nested-PCR for detection, more robust evidence is required to determine whether *Wolbachia* strains are established as endosymbionts in *Anopheles* species [31]. The majority of these studies are limited to amplification of only a few genes (particularly *16S rRNA*) and this is problematic given the possibility of amplifying prokaryotic *16S rRNA* genes from non-living cells[32]”

P3, L59-60

Did this protocol include DNAse treatment?

No unfortunately we did not include DNAase treatment so there is a small possibility of dsDNA being present in the final eluted RNA (despite RNA being the much more abundant nucleic acid extracted due to the kit used). We have added some sentences in the discussion on this.

P4, L5–6

Did you use a Ct threshold after which detection was considered unreliable?

If this is referring to *Wolbachia* quantification (P5 L50 onwards), we used a standard curve using serially diluted synthetic oligonucleotide standards in combination with NTCs. All three technical replicates had to generate a consistent, positive Ct for a sample to be considered 'positive' (Supplementary Figure S1, Supplementary Table S5 for raw qPCR data). Our standard curve and biological sample data (based on consistency of all three technical replicates) resulted in a Ct threshold of ~36. Please note the vast majority of samples were negative (no Ct) and table S5 only provides the qPCR data for samples that we classified as positive (three technical replicate Ct

values).

P4, L44–45

This reads as if the 16S PCR was done on genomic DNA rather than cDNA. Please clarify. **cDNA was used for amplification of 16S and wsp PCR products so we have modified the start of this paragraph for clarity.**

P4, L46–47

The CifA/CifB PCRs are not mentioned anywhere in the results or discussion. Suggest to remove the protocol here.

Apologies this was erroneously included here (amplification was unsuccessful) and we have now removed this.

P5, L38

Please provide justification for using TN model of evolution.

Thank you for highlighting the justification for choice of model. On reflection we have repeated the phylogenetic analyses, with the inclusion of the “Find Best Fit Substitution Model (ML)” option performed within the MEGAX software for each alignment, prior to running the analysis to construct each phylogenetic tree. Each tree has now been constructed using the respective optimal model and options, highlighted through this model selection analysis within the software. Therefore, each tree and the relevant areas of text and figure legends have been updated to reflect this. The re-analysis of the phylogenetic trees has not demonstrated any changes to the results, or required adjustment of the discussion, however, we are glad to now include this additional step for model selection justification and choice optimisation.

P5, L39–40

If this means bootstrap analysis, please provide details here.

The sentence “ The phylogeny test was by Bootstrap method with 1000 replications.” was included further down in this paragraph but this has now been moved up to provide greater clarity here.

P5, L41–44

It appears the description is for the calculation of the starting tree, rather than for the ML reconstructions. Please clarify.

We believe we have included all details and descriptions for the methods of phylogenetic analysis, either those that were generally applicable within the “Phylogenetic analysis” subheading paragraph, or, where specific to each tree, within each individual figure or supplementary figure legend. If any specific details are missing, however, we would be happy to add them.

P5, L49ff

Please provide more details on negative and positive controls here. Did you include Wolbachia-free Anopheles specimens as negative controls?

We optimised this qPCR for onestep qRT PCR from the previously published protocol (Gomes et al. PNAS 2018) using NTCs and synthetic standards (supplementary table 1). We also tested some Wolbachia-infected An. species A (Jeffries et al. WOR 2018) prior to testing our biological samples. The vast majority of *Anopheles* samples in this study were negative (both *An. gambiae* s.s. and *An. melas*) and we only considered samples positive with three consistent technical replicates (supplementary table 5).

P7, L18

It appears there are 2 novel supergroup A strains and 2 belonging to supergroup B. The accession numbers for the B strains seem to be missing from Table S3

Thank you for highlighting this, it appears some rows were missing from this table. The accession numbers for the 16S sequences for both supergroup B strains, as well as *An. sp. X* and *Cx. watti*, have now been added to the table.

P7, L26–50

Is this level of detail really necessary? I think it is almost impossible to infer the relative abundances of two taxa in a sanger chromatogram from a mixed sample. I would suggest to either perform qPCRs for the different strains or to omit these paragraphs.

We agree and have condensed this paragraph to only a few sentences to summarise how repeat 16S sequencing did not help for mixed chromatograms.

P8, L7–21

Again, I think these technical details are not very important. Suggest to consider removing this paragraph.

We agree and have condensed this paragraph to keep only the important information on MLST allelic profiles.

P8, L32

Suggest to change “diverse” to “divergent”.

Thank you for this suggestion and we have done this

P9, L37-38

Please explain why singleplex approaches would lead to reduced sensitivity for the detection of hybrids, this is not immediately apparent

It’s actually the opposite with multiplex approaches resulting in reduced sensitivity: “Our testing highlighted that amplification and clarity of hybrid detection was improved with use of the ribosomal IGS PCR primers[40] for each species in singleplex format, rather than the standard higher-throughput multiplex format, where primers for multiple members of the *An. gambiae* complex are included at the same time, with different product sizes for species discrimination. This is unsurprising due to the designed aims of the multiplex assay, and potential variations in reaction efficiency between species, particularly when hybridised, which were highlighted in the original publication [40].”

P10, L6-7

Unclear what is meant by this statement

All samples that were *Wolbachia* positive by PCR were initially morphologically identified as species within the *An. gambiae* complex. Only through mosquito barcoding assays did we identify this as an ‘unclassified’ *Anopheles* species. We have moved and modified this sentence for greater clarity.

P10, L7–10

Sentence appears incomplete. Suggest to rephrase.

We have reviewed this sentence and it appears complete to us

P10, L21–22

Or rather, the results highlight the necessity for other types of evidence in addition to the molecular data

Yes we agree although the rationale for this sentence is following on from accurate molecular identification of mosquito species rather than confirming the presence of genuine *Wolbachia* strains. We have added a sentence at the end of this paragraph to

show further studies are needed to confirm strains in these species are genuine endosymbionts.